# Assessing Antecedents of Restaurant’s Brand Trust and Brand Loyalty, and Moderating Role of Food Healthiness

**DOI:** 10.3390/nu15245057

**Published:** 2023-12-09

**Authors:** Kyung-A Sun, Joonho Moon

**Affiliations:** 1Department of Tourism Management, Gachon University, Seongnam 13120, Republic of Korea; kasun@gachon.ac.kr; 2Department of Tourism Administration, Kangwon National University, Chuncheon 24341, Republic of Korea

**Keywords:** DINESERV, brand trust, brand loyalty, fast food, Shake Shack, moderating effect

## Abstract

The purpose of this research was to apply DINESERV to a food brand: Shake Shack. Six sub-dimensions (e.g., taste, healthiness, employee service, price fairness, ambience, and convenience) were adopted. This study used brand trust and brand loyalty to explain attributes. This research additionally assessed the moderating impact of healthiness on the relationship between taste and brand loyalty. For data collection, this study used Amazon Mechanical Turk. The main instrument of this research is a survey. The number of valid observations was 353. Confirmatory factor analysis and a correlation matrix were used to ensure the convergent and discriminant validity of measurement items. Structural equation modeling was employed for hypothesis testing. Plus, Hayes process macro model 1 was employed to test the moderating effect of healthiness. Results indicated that brand trust was positively associated with taste (*p* < 0.05), employee service (*p* < 0.05), and ambience (*p* < 0.05), while brand loyalty was positively associated with taste (*p* < 0.05), healthiness (*p* < 0.05), price fairness (*p* < 0.05), ambience (*p* < 0.05), and brand trust (*p* < 0.05). However, the convenience of casual restaurants appeared as a non-significant attribute to account for both brand trust and brand loyalty. The results also revealed that healthiness negatively moderates the relationship between taste and brand loyalty (*p* < 0.05). This study sheds light on the literature by demonstrating the accountability of DINESERV to casual dining customer behavior. Also, this research presents information for the assistance of brand management in the domain of casual dining sector.

## 1. Introduction

Shake Shack has modified the traditional fast food business model. It has accomplished differentiation by focusing on service quality. Shake Shack has adopted a fast casual dining business model and tried to lessen the traditional fast food market’s concern by offering healthier food with organic ingredients such as Angus beef, organic potatoes, and non-frozen meat [1]. Thus, Shake Shack has pursued differentiation, focusing on offering healthy food from traditional fast-food chains such as McDonald’s because traditional fast food is related to health problems such as obesity, high blood pressure, and depression [2,3]. That is, Shake Shack tried to differentiate their market by using popular items (e.g., burgers and shakes) in the area of fast-food restaurants. Thus, it might be valuable to scrutinize the uniqueness of the Shake Shack brand to understand the characteristics of the casual dining business. 

Brand trust and brand loyalty have been commonly investigated by previous studies [4,5,6,7,8,9,10]. Although previous studies have scrutinized revisit intention and satisfaction [11,12,13], scant studies have examined the effect of DINESERV on brand trust and brand loyalty. Considering heavy competition among numerous brands (e.g., McDonald’s, Subway, Pizza hut, Burger King, Taco Bell, Dunkin, Sonic, and Wendy’s) in the food service business domain [14], exploration for brand-related attributes could be worthwhile. 

DINESERV is a framework to assess the service capability of food service businesses from the perspective of consumers using multiple dimensions: food quality, employee service, price fairness, ambience, and convenience [11,12]. Scholars’ evaluations [11,12,13] used five attributes (e.g., food quality, employee service, price fairness, ambience, and convenience) for. Considering both the strengths and weaknesses of fast food, this research will employ two attributes, including taste and healthiness, to measure food quality. Moreover, Kim et al. [11] also proposed five dimensions of DINESERV, which include food quality, employee service, price fairness, ambience, and convenience. By following the approach of Kim et al. [11], this study is to inspect the explanation of DINESERV containing employee service, price fairness, ambience, and convenience. Extant literature has additionally claimed that food quality needs to consider multiple elements and that measurements should be varied by context and consumer expectations [15,16,17,18]. Moreover, scholars alluded to the fact that healthiness has become a more important element in the market because people value their healthy life condition more [19,20,21]. It suggests that consumers are likely to use food healthiness to assess their product quality. Although numerous studies have explored the impact of DINESERV [11,12,13], scholars have also rarely scrutinized food quality attributes using both taste and healthiness. Thus, this research is to adopt food healthiness as a sub-dimension of food quality. Next, scholars alleged that healthiness trades off the effect of taste on consumer appraisal. Namely, consumers assess healthy food as less tasty [22]. It indicated that consumer evaluation of taste is likely to be deterred by a healthy perception of food. Despite such a probability., insufficient studies have been implemented by researchers to test the effect of healthiness on the relationship between taste and brand management. This could become the research gap; this research is to minimize such a research gap in the domain of casual dining restaurants. 

All things considered, this study has the following research questions:

Research question 1: Does DINESERV measuring food quality using taste and healthiness account for brand trust and brand loyalty in the domain of casual restaurants?

Research question 2: Does food healthiness significantly moderate the relationship between taste and brand loyalty?

Therefore, the goal of this research was to determine structural relationships among DINESERV attributes by measuring food quality using two attributes: taste and healthiness, brand trust, and brand loyalty in the casual dining business sector. This study sheds light on the literature by demonstrating the explanatory power of DINESERV for brand management in the casual dining domain. Plus, this research contributes to the literature by ensuring the impact of food healthiness on the association between taste and brand loyalty. Hence, this work is to scrutinize the effect of food healthiness in various ways. Based on the results of this work, this study might be able to provide managers in the casual dining domain with information to allocate their constrained resources for their brand marketing more efficiently. This might have practical implications for this study. 

## 2. Review of Literature and Hypotheses Development

### 2.1. DINESERV

DINESERV refers to the concept that assesses the quality of food service management based on consumption experience [13,23]. Scholars have designated this framework to appraise service quality for food service businesses not only because service varies depending on the context but also because food service businesses are service-oriented businesses [11,12,13]. Specifically, Stevens et al. [23] proposed a tool for measuring food service experience that includes price, ambience, and employee service. Kim et al. [11] have also contemplated food quality as the essence of a dining experience and convenience associated with the accessibility to food because searching for food takes time and effort. Kim and Choi [12] have considered four elements (e.g., employee service, food quality, interior, and price) to assess DINESERV in buffet restaurants. By integrating previous studies through a literature review, this study employs food quality, employee service, price fairness, ambience, and convenience as elements of DINESERV. In sum, DINESERV has demonstrated explanatory power in various food service areas. 

### 2.2. Brand Trust and Brand Loyalty

Brand trust is consumers’ credibility to a certain brand based on the utility of a product [4,7,24]. The extant literature has claimed that brand trust can be assessed by quality and experience in consumption. In the restaurant business domain, brand trust has been investigated as a focal attribute to understand consumers’ psychological mechanism. As an example, Kwon et al. [25] scrutinized determinants of restaurant brand trust. This indicates that brand trust is likely to work as both an independent and dependent variable. Brand loyalty denotes customers’ status to maintain their relationship with certain goods and services as a sort of loyalty behavior [6,26]. Researchers have employed brand loyalty as an outcome variable. Zehir et al. [27] employed brand loyalty as the main attribute, and Lee et al. [28] chose brand loyalty as the main attribute to examine family restaurant customers. Regarding the review of literature, brand loyalty has been widely explored by prior studies.

### 2.3. Relationship between Attributes of DINSERV, Brand Trust and Brand Loyalty

Food quality is the first domain in DINERSERV. Customers consume fast food because it is tasty. Taste is essential for individual decision-making in the food service sector [16,18,29]. Zhong and Moon [30] have demonstrated that taste significantly affects customer loyalty and perception by researching fast food customers. It can be inferred that taste is the critical attribute for consumer appraisal. Additionally, Kim et al. [11] demonstrated that food quality associated with taste exerted a positive impact on consumer decision-making. In addition, Bougoure and Neu [31] showed a positive influence of taste on restaurant customers’ appraisals. Thus, the following research hypotheses are proposed:

**Hypothesis** **1a.**
*Taste is positively associated with brand trust.*


**Hypothesis** **1b.**
*Taste is positively associated with brand loyalty.*


Existing studies have documented concerns about fast food because it harms health. In fact, prior studies have stated that fast food can cause illnesses such as high blood pressure, hyperlipidemia, and diabetes [32,33]. Therefore, healthiness has emerged as a crucial issue in the fast food business domain because people are more concerned about their health condition with improved quality of life [32,34]. Konuk [35] disclosed that food healthiness plays a significant role in building customer trust and loyalty. Additionally, Tian and Kamran [36] performed a systematic review of the literature, and the results indicated that food healthiness is a critical attribute for the decision-making of consumers. Regarding the literature review, this study proposes the following research hypotheses:

**Hypothesis** **2a.**
*Healthiness is positively associated with brand trust.*


**Hypothesis** **2b.**
*Healthiness is positively associated with brand loyalty.*


Previous studies have claimed that restaurant businesses are labor intensive and that service quality can be differentiated depending on service employees’ capabilities [37,38]. Qin et al. [39] and Hansen [40] have revealed that employee service determines perceived quality in the food service area. Zehir et al. [27] scrutinized the structural relationship among service quality, brand trust, and brand loyalty and found that these three attributes are positively associated with each other. Moreover, Alan and Kabadayı [41] have shown that quality positively affects brand trust. Similarly, Jung and Soo [42] reported that brand experience with employee service exerts a positive influence on brand trust. Thus, this research proposes the following research hypotheses:

**Hypothesis** **3a.**
*Employee service is positively associated with brand trust.*


**Hypothesis** **3b.**
*Employee service is positively associated with brand loyalty.*


Scholars argue that the merits of fast food include a rational price [12,43,44,45,46]. This implies that the affordable price of fast food leads customers to prefer fast food because customers perceive a higher level of utility [11,47]. Hride et al. [48] examined e-commerce users, and the results revealed the positive impact of price fairness on trust and loyalty. Chun and Nyam-Ochir [49] revealed that price fairness positively impacted the revisit intentions of fast food restaurants. In a similar vein, Sohaib et al. [50] found the positive influence of price fairness on loyalty by employing hotel customers. By integrating the review of literature, this work proposes the following research hypotheses:

**Hypothesis** **4a.**
*Price fairness is positively associated with brand trust.*


**Hypothesis** **4b.**
*Price fairness is positively associated with brand loyalty.*


Dining conditions such as store interior and cleanliness could become another imperative domain to attract more customers because the store servicescape related to atmosphere and sanitation plays an important role in building customers’ perceptions [51,52,53]. Prior research studies have revealed that ambience is an element that can influence food choice and customer perception [53,54,55]. Chun and Nyam-Ochir [49] revealed that ambience positively affects revisit intentions in fast food restaurants.

**Hypothesis** **5a.**
*Ambience is positively associated with brand trust.*


**Hypothesis** **5b.**
*Ambience is positively associated with brand loyalty.*


Convenience is associated with accessibility because more stores enable consumers to use goods and services more conveniently [11,44]. The likelihood of fast food consumption is increased due to convenience because customers can save their time and effort searching for food [46,56,57]. Prior studies have also documented that convenience is an imperative factor for consumer decision-making because better accessibility leads consumers to appease their hungers more easily [11,12,46]. Considering the arguments of prior literature, this study proposes the following research hypotheses: 

**Hypothesis** **6a.**
*Convenience is positively associated with brand trust.*


**Hypothesis** **6b.**
*Convenience is positively associated with brand loyalty.*


Zehir et al. [27] revealed the positive effect of brand trust on brand loyalty. El Naggar and Bendary [8] disclosed that brand trust has a positive impact on brand loyalty using Egyptian customers. Scholars have also examined the mediating role of brand trust in the relationship between customer perception and brand loyalty [5,9]. Considering the empirical findings of brand loyalty, Hussein [10] has examined the effect of brand experience on brand loyalty by exploring Indonesian restaurant customers. Additionally, Kwon et al. [25] demonstrated that brand trust is positively associated with brand loyalty in the context of restaurant businesses. By considering previous findings, this study has the following research hypothesis:

**Hypothesis** **7.**
*Brand trust is positively associated with brand loyalty.*


### 2.4. Moderating Effect of Healthiness for the Relationship between Taste and Brand Loyalty

Existing literature has indicated that healthiness functions as a cue of less tasty food [22]. Scholars have also claimed that it is a rule of thumb that unhealthy food is tasty, whereas healthy food is less tasty [58,59]. Moreover, Papoutsi et al. [58] claimed that healthiness perception of food reduces the utility of hedonic from taste. Prior studies have also shown that consumers valuing food healthiness are less likely to choose tasty food because of the perception that tasty food is equal to unhealthy food [59,60,61]. This implies that healthiness is likely to offset the effect of taste on consumer perception. In general, consumers’ purchase decisions are likely to be affected by food taste [11]. It is presumed that healthiness is likely to exert a lower magnitude of the impact of taste on brand loyalty. Namely, the utility of taste is likely to be diminished by the healthy frame. Regarding the review of literature, this research proposes the following research hypothesis:

**Hypothesis** **8.**
*Healthiness exerts a significant moderating effect on the relationship between taste and brand loyalty.*


## 3. Methods

### 3.1. Research Model

Figure 1 describes the research model. DINESERV possesses six elements: taste, healthiness, employee service, price fairness, ambience, and convenience. DINESERV attributes account for brand trust and brand loyalty. Taste, healthiness, employee service, price fairness, ambience, and convenience can positively affect both brand trust and brand loyalty. Additionally, brand trust can positively affect brand loyalty. 

Figure 2 is the second research model for this work. Healthiness is the moderating variable in the relationship between taste and brand loyalty. Taste is the independent variable, while brand loyalty is the dependent variable.

### 3.2. Measurements and Data Collection

Survey is the main instrument. To measure the main attributes, a five-point Likert scale (1 = strongly disagree, 5 = strongly agree) was employed. These measurement items were derived from prior studies. They were adjusted based on the purpose of the research. Table 1 displays these measurement items. The definition of taste is related to consumers’ assessment of whether a company’s food is delicious [18,29,30]. The notion of healthiness is about whether a company’s food promotes health conditions [32,33,34]. This study defined employee service as the evaluation of the attitude of food service that a company provides [11,38,49]. Price fairness is the perception about whether the price is affordable for the food product of a company [11,12,46,48]. The definition of ambience is about how customers perceive cleanliness and comfort using fast food stores [11,53,54,55]. In this research, the notion of convenience is how consumers easily obtain goods and services from the food service business [11,46,57]. The definition of brand trust is about the degree of credibility for a food service business brand [4,7,24]. Last, the concept of brand loyalty concerns how customers intend to behave for sales growth in food service businesses [6,25,28].

This study measured the main attributes using the actual brand name, Shake Shack. Four items were employed to measure taste, healthiness, price fairness, brand trust, and brand loyalty. Employee service and convenience were measured using three items. Ambience measurements were performed using five items. Demographic characteristics included age (1 = 20–29 years old or younger, 2 = 30–39 years old, 3 = 40–49 years old, 4 = 50–59 years old, 5 = older than 60 years old), gender (0 = male, 1 = female), monthly household income (1 = less than USD 2000, 2 = USD 2000–3999, 3 = USD 4000–5999, 4 = USD 6000–7999, 5 = USD 8000–9999, 6 = more than USD 10,000), and monthly visiting frequency (1 = less than 1 time, 2 = 1–2 times, 3 = 3–5 times, 4 = more than 5 times).

Survey responses were obtained using Amazon Mechanical Turk. Amazon Mechanical Turk holds abundant American panels for survey responses. Scholars have selected Amazon Mechanical Turk to collect data. It offers guaranteed quality of data for statistical inference [62,63,64]. The study type of this research is a cross-sectional survey that collected data from numerous study participants [65]. This research chose Shake Shack as the research domain because of their popularity in the American food service market. Such popularity can lead survey participants to respond, considering their actual experience. Plus, the use of Amazon Mechanical Turk, which has numerous American-based survey participants, might be suitable because Shake Shack is an American-based food service business. Also, the renowned brand might provide more specific context to assess brand trust and brand loyalty. The period of data collection was between 12 February 2021 and 16 February 2021. The number of original observations was 452. This study initially asked whether participants had experienced Shake Shack’s goods and services. Non-experienced survey participants were excluded for data analysis. After eliminating 99 observations, the final number of observations was 353 for data analysis. Plus, this study did not collect sensitive personal information such as birthdays, social numbers, private behavior, or contact information. By doing so, this study tried to protect survey participants’ privacy. 

### 3.3. Data Analysis

At the beginning, this research carried out frequency analysis for demographic information. This research also executed an analysis of variance to compare differences in eight main attributes (e.g., taste, healthiness, employee service, price fairness, ambience, convenience, brand trust, and brand loyalty) with attributes of demographic information to explore the information in the data more. Tukey was used as an instrument for post-hoc analysis. This research then implemented confirmatory factor analysis, a correlation matrix, and path analysis using a structural equation model. To ensure convergent validity, factor loading with a threshold of 0.5 and a construct reliability cut-off value of 0.7, which ensures the reliability of measurement items, was examined [66,67]. Plus, the rule that average variance extracted (AVE) is greater than 0.5 to assess the discriminant validity of measurement items is likely to cause bias by multicollinearity [68]. This research also employed the instruction that the square root of AVE should be greater than the correlation coefficient for testing discriminant validity [67,69]. 

In order to test hypotheses, a structural equation model was performed. Considering existing literature, the goodness of fit of the structural equation model was attested with multiple indices, including Q (CMIN/degree of freedom) < 3, RMR (root mean-square residual) < 0.05, RMSEA (root mean square error of approximation) < 0.05, GFI (goodness of fit index) > 0.8, NFI (normed fit index) > 0.8, RFI (relative fit index) > 0.8, IFI (incremental fit index) > 0.8, TLI (Tucker–Lewis Index) > 0.8, and CFI (comparative fit index) > 0.8 [69,70]. This study used analysis of moment structure (AMOS) as an instrument for structural equation modeling and confirmatory factor analysis. Moreover, this research used a maximum likelihood-based structural equation model because it requires more than 250 observations for significant statistical inference [68,69].

Next, this work employed Hayes process macro model 1 with 5000 bootstrapping to attest to the moderating effect of healthiness on the relationship between taste and brand loyalty. Hayes process macro model 1 is to examine the effect of moderating variables (W) on the relationship between independent variables (X) and dependent variables (Y). The process macro model is an ordinary least square-based analytic instrument, and the estimation is less likely to be biased because it does not assume normality of the data for parameter estimation [69]. The statistical package for the analysis is process macro from the statistical package in social science (SPSS) 21.0 version. In order to scrutinize the moderating effect, this research performed a median split analysis to divide the group. The median values of taste and healthiness are 4.25 and 3.25, respectively. Using four groups, this research computed the mean value of brand loyalty for each group.

## 4. Results

### 4.1. Profile of Survey Participants

Table 2 shows information about survey participants. There were 199 male participants and 154 female ones. Table 2 describes their demographic information for age (20–29 years old or younger: 83, 30–39 years old: 174, 40–49 years old: 65, 50–59 years old: 18, and older than 60 years old: 13) and monthly household income (less than USD 2000: 70; between USD 2000 and USD 3999: 117; between USD 4000 and USD 5999: 82; between USD 6000 and USD 7999: 37; between USD 8000 and USD 9999: 19; and more than USD 10,000: 28). Regarding monthly visiting frequency, it was less than 1 time for 125 participants, 1–2 times for 165, 3–5 times for 52, and more than 5 times for 11.

### 4.2. Confirmatory Factor Analysis and Correlation Matrix

Table 3 depicts the results of the confirmatory factor analysis. Seven attributes were examined. The goodness of fit indices denoted the significance of the results. Considering factor loading and construct reliability, values were greater than cut-off values, confirming the convergent validity of measurements. Although the criteria for RMSEA is 0.05, existing literature suggests that 0.09 could be considered an acceptable level [68,69]. Table 3 also presents information on the mean values and average variance extracted. Plus, the average variance extracted met the criteria. 

Table 4 presents a correlation matrix. First, brand loyalty was positively correlated with taste (r = 0.791, *p* < 0.05), healthiness (r = 0.562, *p* < 0.05), employee service (r = 0.741, *p* < 0.05), price fairness (r = 0.693, *p* < 0.05), ambience (r = 0.796, *p* < 0.05), convenience (r = 0.695, *p* < 0.05), and brand trust (r = 0.885, *p* < 0.05). Moreover, correlation coefficients were smaller than the square root of the average variance extracted, indicating acceptable discriminant validity in both cases. In addition, the correlation coefficient between brand trust and brand loyalty is greater than the diagonal values. However, the likelihood of multi-collinearity in estimation is very low because both variables are in the causal relationship in the hypothesis.

### 4.3. Results of Hypotheses Testing Using Structural Equation Model

Table 5 shows the results of hypothesis testing. Brand trust was positively influenced by taste (β = 0.442, *p* < 0.05), employee service (β = 0.129, *p* < 0.05), and ambience (β = 0.176, *p* < 0.05); taste (β = 0.190, *p* < 0.05), healthiness (β = 0.129, *p* < 0.05), price fairness (β = 0.110, *p* < 0.05), ambience (β = 0.176, *p* < 0.05), and brand trust (β = 0.488, *p* < 0.05) exerted a positive effect on brand loyalty. In summary, H1a, H1b, H2b, H3a, H4b, H5a, H5b, and H7 were supported in the Shake Shack.

### 4.4. Results of Hypotheses Testing Using Structural Equation Model

Table 6 shows the results of the moderating effect using Hayes process macro model 1. The results disclosed that taste is positively associated with brand loyalty (β = 0.868, *p* < 0.05). In addition, the results show that taste × healthiness is negatively associated with brand loyalty (β = −0.073, *p* < 0.05). It implied that the effect of taste on brand loyalty could be lowered by the moderating impact of healthiness. The results are statistically significant based on the F-value (*p* < 0.05). All values of the conditional effect of healthiness are significant, and the magnitude becomes lower (β (Taste_Healthiness 2_ → brand loyalty) = 0.7214, β (Taste_Healthiness 3.25_ → brand loyalty) = 0.6291, and β (Taste_Healthiness 4.25_ → brand loyalty) = 0.5554). The test of unconditional interaction indicates that the interaction exerts a statistically significant effect to account for brand loyalty, considering the F-value (*p* < 0.05). The change in R-square (0.0058) noted the increased explanatory power of the moderating variable.

Figure 3 presents the results of the moderating effect of healthiness on the association between taste and brand loyalty. The mean of the low taste and low healthiness groups is 3.07, and the mean of the high healthiness and high taste groups is 4.45. In addition, Figure 3 and Table 7 document the mean values of the low taste and high healthiness groups (mean = 3.79) and the high taste and low healthiness groups (mean = 4.03), respectively. 

The results of the analysis of variance are presented in Table 8. For gender, healthiness showed a significant difference (Mean_male_ = 3.301 and Mean_female_ = 2.964). Regarding monthly using frequency, taste (Mean_less than 1_ = 3.812, Mean_1–2_ = 4.290, Mean_3–5_ = 4.365, and Mean_more than 5_ = 4.681), healthiness (Mean_less than 1_ = 2.600, Mean_1–2_ = 3.333, Mean_3–5_ = 3.716, and Mean_more than 5_ = 4.045), employee service (Mean_less than 1_ = 3.718, Mean_1–2_ = 4.066, Mean_3–5_ = 4.177, and Mean_more than 5_ = 4.545), price fairness (Mean_less than 1_ = 3.518, Mean_1–2_ = 3.897, Mean_3–5_ = 4.033, and Mean_more than 5_ = 4.340), ambience (Mean_less than 1_ = 3.690, Mean_1–2_ = 4.054, Mean_3–5_ = 4.187, and Mean_more than 5_ = 4.590), convenience (Mean_less than 1_ = 3.316, Mean_1–2_ = 3.783, Mean_3–5_ = 4.019, and Mean_more than 5_ = 4.522), brand trust (Mean_less than 1_ = 3.654, Mean_1–2_ = 4.107, Mean_3–5_ = 4.206, and Mean_more than 5_ = 4.727), and brand loyalty (Mean_less than 1_ = 3.374, Mean_1–2_ = 4.025, Mean_3–5_ = 4.230, and Mean_more than 5_ = 4.613) revealed statistically significant difference. It indicates that monthly usage frequency is likely to be influenced by the attributes of DINESERV, brand trust, and brand loyalty.

## 5. Discussion

The results showed that taste resulted in a higher level of brand trust and loyalty. Healthiness also appeared to be a remarkable element in making customers more loyal to the Shake Shack brand. For healthiness, the mean values are lower than for other attributes. It might be due to the business characteristics of selling hamburgers and shakes, which could be regarded as representative fast food due to their high calories. In addition, the fast-food image of hamburgers might be limited to building brand trust by appealing to healthiness from the perspective of consumers. Furthermore, it could be inferred that employee service exerted a positive influence on brand trust. However, employee service was not influential on brand loyalty. This could be due to the fact that the interaction with employees is not active because the role of employees tends to be limited to just getting orders in the casual dining sector as compared to the full-service restaurant sector. Next, the price fairness of Shake Shack enhanced the level of brand loyalty. Results for Shake Shack also demonstrated that ambience encouraged customers to become both more credible and loyal to the brand in the area of casual dining restaurant business. Plus, results disclosed a positive impact of brand trust on brand loyalty at Shake Shack. Additionally, the results indicated that brand trust is positively associated with brand loyalty in the case of casual dining customers. It ensured the findings of prior studies [9,10,25]. Plus, the results revealed that convenience was not a significant element for either brand trust or brand loyalty. The outcomes of this work externally validated the findings of the extant literature [10,25,28]. Furthermore, the results of this work revealed the significant negative effect of healthiness on the impact of taste on brand loyalty of casual dining. That is, the impact of taste could be decreased in the case of a high-healthiness group to build brand loyalty among casual dining restaurant consumers. It empirically demonstrated the argument that healthiness and taste are likely to become the tradeoff relationship from previous studies by using brand loyalty as the main attribute [61,62,63].

In general, the results of this research aligned with the findings of the extant literature employing consumers from other domains, such as fast food and institutional food service, by demonstrating the significant effect of food quality, employee service, price fairness, and ambience on consumer behavior [11,32,49]. That is, this research found influential attributes on brand management attributes in the area of casual dining restaurant businesses. This might be due to the number of stores, which constrains the accessibility of the Shake Shack product compared to other hamburger and sandwich brands: McDonald’s Burger King, Wendy’s and others. That is, the limited accessibility could explain the nonsignificance because the number of casual dining stores is relatively limited compared to the number of dining facilities in schools from the viewpoint of university students, which is the finding of Kim et al. [11].

## 6. Conclusions

### 6.1. Theoretical Contributions

This study theoretically contributes to the literature. First, this research ensured the accountability of DINESERV in the domain of casual dining businesses. It not only validated the results of Bougoure and Neu [31] and Chun and Nyam-Ochir [49] but also expanded the area of DINESERV literature to the fast casual dining sector. Second, prior studies have claimed that the single measure of food quality is limited and that the definition is different depending on the context and customer expectation [15,17,68]. Given this argument, this study scrutinized subdimensions (e.g., taste and healthiness) of food quality contemplating fast food characteristics. Such measures allowed this study to perform more in-depth statistical inference. That is, this study sheds light on the literature by proposing an advanced DINESERV model and demonstrating its accountability in the context of casual dining restaurant customer management. In addition, previous studies have commonly adopted revisit intention and satisfaction as explained variables [11,49]. It can be inferred that it could be valuable to adopt more diverse explained attributes. Moreover, multiple brands (e.g., McDonald’s, Burger King, Subway, Shake Shack, Wendy’s, In and Out, etc.) are competing in the food service business market. Considering both the research gap and market conditions, this study employed brand trust and brand loyalty as outcome variables and confirmed the accountability of DINESERV for these two attributes. By doing so, this study contributes to the literature by providing a deeper understanding of consumers’ behavior in the area of casual restaurant domain regarding brand management as the central piece. Furthermore, this research sheds light on the literature by elucidating the moderating effect of healthiness on the association between taste and brand loyalty. This could become the validation of the results of the extant literature in the area of food service brand management [58,70]. Last, this research theoretically contributes to the literature by empirically disclosing scholars’ claim that healthiness has a tradeoff with the effect of taste on individual behavior [58,59,60,61]. The results of the moderating effect of healthiness on the relationship between taste and brand loyalty could become evidence.

### 6.2. Managerial Implications

This study suggests the following managerial implications: For fine casual dining, food healthiness appeal might need to be accomplished by communicating healthy food ingredients (e.g., fresh vegetables and beef) to customers. For employee management, only casual dining businesses like Shake Shack need to spend their budget on employee training for service attitude and capability. Moreover, managers of casual dining restaurants might need to concentrate more on taste because it can enhance both brand trust and loyalty. Better taste might be accomplished by the employee training for cooking manuals in kitchen staff as well as the supplier management because they provided fresh and tasty ingredients for food. Namely, food service managers need to invest their resources more in the management of kitchen staff training and supplier administration. In addition, casual dining restaurant managers might be able to approach their pricing strategy more parsimoniously because price variation might undermine the perception of price fairness, which might result in losing loyal customers. Plus, casual restaurant managers might need to allocate their resources more toward the ambience of the store. It might be achieved by furnishing the store with nice-looking decoration and interiors and maintaining clean store conditions. Such an effort might lead customers to assess the brand of a casual dining restaurant better, which is associated with competitive advantage in business. Finally, casual dining restaurant managers might dedicate their budget more to building brand trust, which affects brand loyalty. Brand trust might be upgraded by environmental, social, and governance (ESG) implementation. In other words, casual dining restaurant managers might be able to contemplate executing more ESG for a better brand reputation for their business. Plus, managers in the area of casual dining restaurants might be able to invest in healthy products because consumers are likely to appraise healthy food as less tasty food. Because taste is an essential element in the food service domain, marketing managers might need to prepare ways to minimize the suspicions of consumers in the area of casual dining service. However, casual dining restaurant managers might need to be more cautious when launching a healthy menu because healthiness could lower the utility of the taste for customer loyalty. In other words, restaurant managers might need to contemplate healthy menu planning because it could lead to inefficient resource allocation.

### 6.3. Suggestions for Future Research

This study has some limitations. First, this study chose single brands to figure out consumer characteristics in the food service business domain. Future research needs to consider more diverse restaurant sectors in either the internal or external domain of casual restaurant business to obtain more valuable outcomes because the unique point of a restaurant brand is likely to be varied. Future research also needs to adopt more diverse attributes (e.g., purchase intention, switching intention, and willingness to pay a premium) as well as methods (e.g., observational and experimental methods) to explore customer behavior and individual decision making. In addition, future research should use more diverse attributes to appraise and measure food quality. Such work will make the area of food service business studies more fruitful. Furthermore, future research might be able to consider the effect of healthiness on consumer behavior in a more diverse cultural context because food healthiness could be defined in diverse ways depending on the context.

## Figures and Tables

**Figure 1 nutrients-15-05057-f001:**
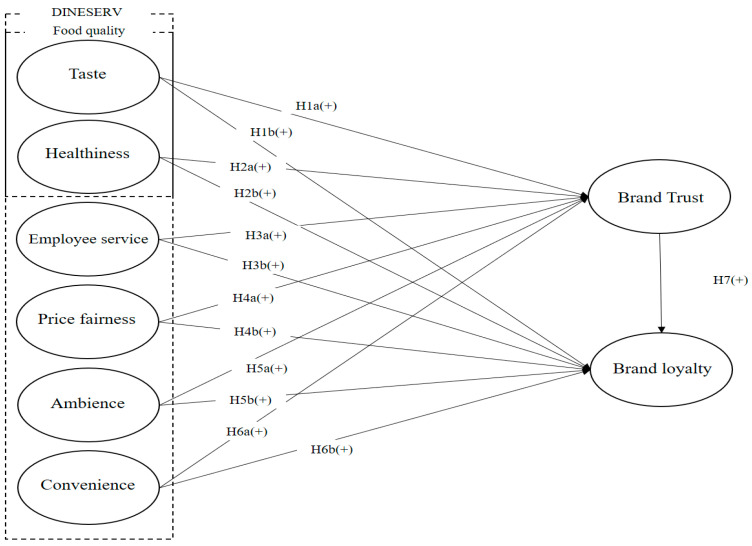
Relationship DINESERV attributes, brand trust, and brand loyalty.

**Figure 2 nutrients-15-05057-f002:**
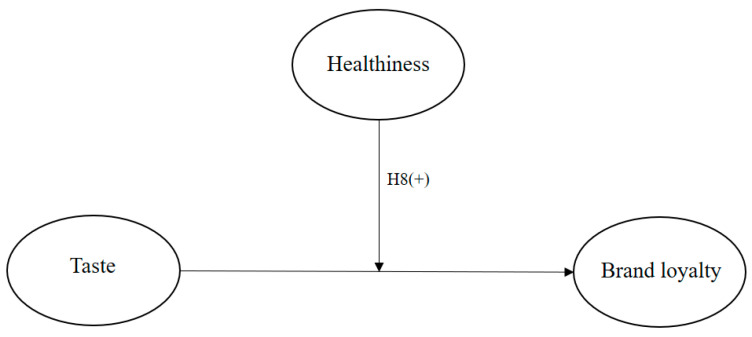
Moderating effect of healthiness on the impact of taste on brand loyalty.

**Figure 3 nutrients-15-05057-f003:**
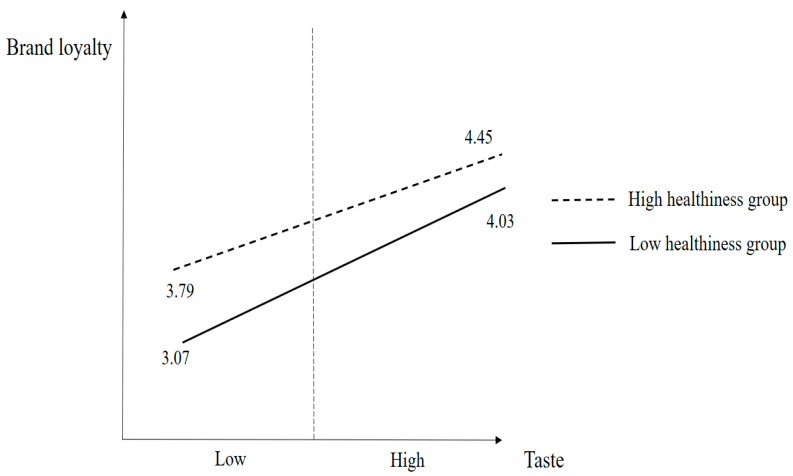
Moderating effect of healthiness on the relationship between taste and brand loyalty.

**Table 1 nutrients-15-05057-t001:** Description of measurements.

Construct	Code	Item
Taste	TA1	Shake Shack burger is tasty.
	TA2	Shake Shack menu is delicious.
	TA3	Shake Shack food is flavorful.
	TA4	Shake Shack offers tasty food.
Healthiness	HE1	Shake Shack food is healthy.
	HE2	Shake Shack food is nutritional.
	HE3	Shake Shack food promotes health condition.
	HE4	Shake Shack food contains healthy ingredient.
Employee service	ES1	Shake Shack employees are kind.
	ES2	Shake Shack employees are cooperative.
	ES3	Shake Shack employees are helpful.
Price fairness	PF1	Price of Shake Shack product is fair.
	PF2	Price of Shake Shack product is rational.
	PF3	Shake Shack offers acceptable price level.
	PF4	Shake Shack product price is reasonable.
Ambience	AM1	Shake Shack store is clean.
	AM2	Shake Shack store cleanliness is administered well.
	AM3	Shake Shack offers comfort dining ambience.
	AM4	Shake Shack provides restful dining condition.
	AM5	Shake Shack store is cozy.
Convenience	CO1	Shake Shack store is accessible.
	CO2	Shake Shack product is easy to reach.
	CO3	Shake Shack is convenient to visit.
Brand trust	BT1	Shake Shack is credible brand.
	BT2	Shake Shack brand is trustworthy.
	BT3	Shake Shack is reliable brand.
	BT4	I trust Shake Shack brand name.
Brand loyalty	BL1	I continue to use Shake Shack brand.
	BL2	I am loyal to Shake Shack brand.
	BL3	Shake Shack brand deserves to purchase again.
	BL4	I would recommend Shake Shack brand to others.

**Table 2 nutrients-15-05057-t002:** Demographic information of participants.

Item	Frequency (%)
Male	199(56.4)
Female	154(43.6)
20–29 years old or younger	83(23.5)
30–39 years old	174(49.3)
40–49 years old	65(18.4)
50–59 years old	18(5.1)
Older than 60 years old	13(3.7)
Monthly household income	
Less than $2000	70(19.8)
Between $2000 and $3999	117(33.1)
Between $4000 and $5999	82(23.2)
Between $6000 and $7999	37(10.5)
Between $8000 and $9999	19(5.4)
More than $10,000	28(7.9)
Monthly visiting frequency	
Less than 1 time	125(35.4)
1~2 times	165(46.7)
3~5 times	52(14.7)
More than 5 times	11(3.1)
Total	353(100.0)

**Table 3 nutrients-15-05057-t003:** Results of confirmatory factor analysis.

Construct (AVE)	Code	Mean	Loading	Critical Ratio	Construct Reliability
Taste (0.707)	TA1	4.14	0.859		0.906
	TA2	4.11	0.835	19.657 *	
	TA3	4.12	0.815	18.878 *	
	TA4	4.19	0.854	20.385 *	
Healthiness (0.784)	HE1	3.08	0.877		0.784
	HE2	3.20	0.880	23.070 *	
	HE3	3.06	0.901	24.208 *	
	HE4	3.26	0.884	23.304 *	
Employee service (0.717)	ES1	4.01	0.844		0.884
	ES2	4.09	0.849	19.147 *	
	ES3	4.11	0.848	19.123 *	
Price fairness (0.635)	PF1	3.78	0.780		0.874
	PF2	3.81	0.716	13.798 *	
	PF3	3.78	0.832	16.422 *	
	PF4	3.81	0.853	16.886 *	
Ambience (0.597)	AM1	4.07	0.761		0.881
	AM2	4.03	0.739	14.163 *	
	AM3	3.85	0.792	15.326 *	
	AM4	3.87	0.811	15.752 *	
	AM5	3.69	0.757	14.546 *	
Convenience (0.674)	CO1	3.94	0.839		0.861
	CO2	3.83	0.842	18.011 *	
	CO3	3.64	0.780	16.343 *	
Brand trust (0.692)	BT1	3.96	0.802		0.900
	BT2	3.95	0.836	17.918 *	
	BT3	4.01	0.852	18.407 *	
	BT4	3.98	0.836	17.923 *	
Brand loyalty (0.672)	BL1	3.97	0.815		0.891
	BL2	3.53	0.754	15.909 *	
	BL3	3.95	0.863	19.244 *	
	BL4	3.91	0.843	18.605 *	

Note: AVE stands for average variance extracted, *, *p* < 0.05. Goodness of fit indices: χ^2^ = 901.623 df = 406, Q(χ^2^/df) = 2.221 RMR = 0.042, GFI = 0.853, NFI = 0.901, RFI = 0.887, IFI = 0.943, TLI = 0.934, CFI = 0.943 RMSEA = 0.059.

**Table 4 nutrients-15-05057-t004:** Correlation matrix.

	1	2	3	4	5	6	7	8
1. Taste	0.841							
2. Healthiness	0.395 *	0.885						
3. Employee service	0.635 *	0.385 *	0.847					
4. Price fairness	0.573 *	0.495 *	0.652 *	0.797				
5. Ambience	0.667 *	0.552 *	0.791 *	0.666 *	0.773			
6. Convenience	0.646 *	0.485 *	0.698 *	0.680 *	0.760 *	0.821		
7. Brand trust	0.798 *	0.460 *	0.770 *	0.644 *	0.768 *	0.687 *	0.832	
8. Brand loyalty	0.791 *	0.562 *	0.741 *	0.693 *	0.796 *	0.695 *	0.885 *	0.820

Note: *, *p* < 0.05, diagonal-square root of AVE.

**Table 5 nutrients-15-05057-t005:** Results of path analysis.

Hypothesis	Path	β(t-Value)	Results
H1a	Taste → Brand trust	0.442(7.79) *	Supported
H1b	Taste → Brand loyalty	0.190(3.10) *	Supported
H2a	Healthiness → Brand trust	0.045(1.01)	Not supported
H2b	Healthiness → Brand loyalty	0.129(3.19) *	Supported
H3a	Employee service → Brand trust	0.293(3.96) *	Supported
H3b	Employee service → Brand loyalty	0.022(0.31)	Not supported
H4a	Price fairness → Brand trust	0.061(1.05)	Not supported
H4b	Price fairness → Brand loyalty	0.110(2.11) *	Supported
H5a	Ambience → Brand trust	0.176(2.04) *	Supported
H5b	Ambience → Brand loyalty	0.162(2.05) *	Supported
H6a	Convenience → Brand trust	0.001(0.01)	Not supported
H6b	Convenience → Brand loyalty	−0.038(−0.60)	Not supported
H7	Brand trust → Brand loyalty	0.488(5.82) *	Supported

Note: *, *p* < 0.05.

**Table 6 nutrients-15-05057-t006:** Results of Hayes process macro model 1.

Variable	β(t-Value)
Constant	−0.559(−1.43)
Taste	0.868(9.35) *
Healthiness	0.567(3.83) *
Taste × Healthiness	−0.073(−2.20) *
F-value	163.78 *
R-square	0.5847 *
Conditional effect of healthiness	
Healthiness (2)	0.7214(15.59) *
Healthiness (3.25)	0.6291(12.25) *
Healthiness (4.25)	0.5554(7.45) *
Test of unconditional interaction	
F-value	4.86 *
R-square change	0.0058

Note: *, *p* < 0.05, dependent variable is brand loyalty.

**Table 7 nutrients-15-05057-t007:** Results of median split analysis.

Group	Low Taste	High Taste
High healthines	3.79	4.45
Low healthiness	3.07	4.03

**Table 8 nutrients-15-05057-t008:** Results of analysis of variance.

Group	Gender	Age	Monthly Household Income	Monthly Using Frequency
Taste	0.021	1.543	0.399	15.700 *
Healthiness	7.845 *	0.724	2.715	21.027 *
Employee service	1.053	0.207	0.596	10.904 *
Price fairness	1.866	0.642	0.118	11.536 *
Ambience	0.596	0.797	0.431	11.575 *
Convenience	0.945	0.414	2.963	16.341 *
Brand trust	0.129	1.058	0.253	16.072 *
Brand loyalty	0.996	0.095	0.475	24.713 *

Note: Values in the table are F-statistics, * *p* < 0.01.

## Data Availability

Data are contained within the article.

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
