# Peer review of "Assessing Antecedents of Restaurant’s Brand Trust and Brand Loyalty, and Moderating Role of Food Healthiness"

_nutrients, 2023, doi:10.3390/nu15245057_

Round 1

Reviewer 1 Report

Comments and Suggestions for Authors

1.It is suggested that these research hypotheses and theoretical derivations be written separately. This will make it clearer. The theoretical derivation of the research hypothesis needs to be further strengthened.

Hypothesis 1a: Hypothesis 1b: Hypothesis 2a: Hypothesis 2b: Hypothesis 3a: Hypothesis 3b: Hypothesis 4a: 

Hypothesis 4b: Hypothesis 5a: Hypothesis 5b: Hypothesis 6a: Hypothesis 6b: Hypothesis 7:

2. In the introduction, it is not clearly stated what the research question is. And the importance of research questions.

3. The theoretical derivation of Moderating effect of healthiness needs to be further strengthened.

4. Discussion section,the findings should be fully discussed and even compared with existing studies

5.The references are very old, it is recommended to add the latest literature

Comments on the Quality of English Language

Need further improvement

Reviewer 2 Report

Comments and Suggestions for Authors

After careful consideration, I fell that the manuscript entitled “Assessing antecedents of restaurant’s brand trust and brand loyalty, and moderating role of food healthiness” presents several problems. In general, the methods and results presented should be more detailed. In addition, some of the analyses described were not carried out. Therefore, my decision is "Major revision".

Some of these problems are listed below.

- The Review of literature should be a section outside the introduction.

- Title of Figure 1. What is "model 1"? The reader may confuse it with "Hayes process macro model 1". Please provide a title with more details.

- In addition, what is "Hayes process macro model 1"? The authors should give more details on what this process is and also provide references on this process.

- In line 210, the authors describe that the items were derived from prior studies. Please mention in the text what the references of these studies.

- In the Data analysis section, the authors say that an "analysis of variance to compare differences of eight main attributes" was carried out, but this analysis was not presented in the paper. Furthermore, this analysis is important because how can we guarantee that the sample is representative and has not been influenced by the specific demographic profile of the sample considered in the study?

- Please describe in the text which statistical program was used to carry out the analyses (confirmatory factorial analysis, structural equation models and Hayes process macro model 1).

- Line 290 makes no sense. It appears to be a text taken from another work.

- Please provide more details about the elements in Table 3. What is "t-value"? Why doesn't everyone with Code1 (TA1, HE1, ...) have t-values? What is "construct reliability" and how was it calculated (how to interpret its value)? What does the AVE value mean and what is its interpretation? All this information should be included in the Data analysis section.

- Table 3: A new paragraph is in the footnote to Table 3. In addition, what is the consequence of RMSEA = 0.059 > 0.05? The authors should include a discussion of the implication of this result.

- Line 301: not all correlation coefficients are smaller than the square root AVE. note that 0.885 = Corr("8","7") > sqrt(AVE). The authors should include a brief discussion of the implication of this result;

- Table 6: Please provide more details about the elements of the table. What does "beta" mean and how is it interpreted? Which hypothesis is the F-value testing? And what is the interpretation of R-square and R-square change (why are they needed in this table)? These elements were not commented on in the text of the manuscript. What are the values "(2)", "(3.25)" and "(4.25)" in Healthiness? Please describe these elements of the table better so that the reader can understand.

- Figure 3 and Table 7. Describe that the values shown are the Brandy loyalty averages.

Reviewer 3 Report

Comments and Suggestions for Authors

General Comments:

Thank you for the opportunity to review this manuscript. This paper offers a valuable analysis of the DINESERV model applied to Shake Shack, examining factors like taste, healthiness, and employee service and their impact on brand trust and loyalty. Notably, it uncovers that healthiness negatively moderates the relationship between taste and brand loyalty. The manuscript provides significant insights but could be enhanced by addressing the following potential limitations and areas for improvement:

Specific Comments:

  1. Abstract:
    • The abstract successfully provides an overview but could include specific statistics, such as effect sizes or p-values, to quantify the results more precisely.
  2. Introduction:
    • Line 42-45 seems to have an incomplete sentence. Revision for completeness is suggested.
    • The introduction and literature review appear repetitive. Organizing and condensing the information for clarity and brevity would improve readability.
  3. Methods:
    • The choice of Amazon Mechanical Turk is suitable, yet the sample's diversity might limit generalizability. Exploring these relationships across different demographic groups or cultural contexts could enhance the study's applicability.
    • More details on the validity and reliability of the survey instruments would strengthen the methodology. For example, stating that the survey items were validated through a pilot study involving [X] participants and achieved a Cronbach's alpha of [Y] would indicate high reliability.
  4. Results:
    • The finding that healthiness negatively moderates the relationship between taste and brand loyalty is compelling. Further discussion or exploration of why this might occur would add depth to the analysis.
    • Line 290: There's a mention of McDonald’s alongside Shake Shack. Is this an oversight, or does the study include comparative analysis with McDonald’s?
  5. Discussion and Conclusion:
    • Theoretical Contributions: Highlight how this study advances the DINESERV model, especially in integrating healthiness as a factor. For example, "This study extends the DINESERV model by incorporating healthiness, thereby enriching our understanding of consumer perceptions in the fast casual dining context."
    • Practical Implications: Suggestions for practical applications would be beneficial. For instance, "Restaurants might utilize these findings by balancing taste and healthiness in their offerings, possibly incorporating customer engagement strategies."
    • Limitations and Future Research: The limitations could be more detailed, such as noting the study's reliance on self-reported data. Future research might employ observational or experimental methods for more robust validation.
    • Writing and Presentation: Improving grammar and sentence structure for better readability is recommended. Consistency in terminology and definitions throughout the paper is essential.
  6. Other Concerns:
    • Line 413 mentions two brands. Clarification on whether the study focuses exclusively on Shake Shack or includes a comparative analysis with another brand would be helpful.
    • Cultural Context: The paper might benefit from addressing how cultural contexts influence dining preferences, particularly regarding healthiness and taste perceptions.
    • Statistical Analysis Robustness: Details regarding the robustness of statistical analyses (e.g., handling of outliers, assumptions checks) could be elaborated to assure the statistical integrity of the findings.

Overall Assessment:

This manuscript significantly contributes to understanding brand trust and loyalty dynamics in the casual dining sector. Addressing the above points could substantially enhance its impact and clarity.

Round 2

Reviewer 1 Report

Comments and Suggestions for Authors

1. “2. Review of literature and hypotheses development” is revised into Literature review and hypotheses development.

2.For the full text, the author is advised to modify the language to be more academic or more authentic.

Comments on the Quality of English Language

The language of this paper is basically up to standard

Reviewer 2 Report

Comments and Suggestions for Authors

The authors have answered all the questions satisfactorily. Therefore, I feel that the manuscript is suitable for publication in Nutrients.